# In Vitro Organoid-Based Assays Reveal SMAD4 Tumor-Suppressive Mechanisms for Serrated Colorectal Cancer Invasion

**DOI:** 10.3390/cancers15245820

**Published:** 2023-12-13

**Authors:** Kevin Tong, Manisha Bandari, Jillian N. Carrick, Anastasia Zenkevich, Om A. Kothari, Eman Shamshad, Katarina Stefanik, Katherine S. Haro, Ansu O. Perekatt, Michael P. Verzi

**Affiliations:** 1Department of Genetics, Rutgers, The State University of New Jersey, Piscataway, NJ 08854, USAaperekat@stevens.edu (A.O.P.);; 2Human Genetics Institute of New Jersey, Piscataway, NJ 08854, USA; 3Cancer Institute of New Jersey, New Brunswick, NJ 08901, USA; 4Hackensack Meridian Health Center for Discovery and Innovation, Nutley, NJ 07110, USA; 5Department of Medical Sciences, Hackensack Meridian Health School of Medicine, Nutley, NJ 07110, USA; 6Department of Biology, The College of New Jersey, Ewing Township, NJ 08618, USA; 7Department of Chemistry and Chemical Biology, Stevens Institute of Technology, Hoboken, NJ 07030, USA; 8Rutgers Center for Lipid Research, New Brunswick, NJ 08901, USA

**Keywords:** colon cancer, invasion, organoids, SMAD4

## Abstract

**Simple Summary:**

BRAF-driven serrated colon cancers are among the most aggressive subsets of colorectal cancers. Smad4 is a critical tumor suppressor found mutated in many late-stage cancers, yet the role of Smad4 in serrated cancer invasion is still relatively understudied. This study seeks to address this gap by using organoids as an in vitro system for invasive behavior. Organoids derived from BRAF-driven tumors have invasive capabilities, and the re-expression of SMAD4 directly suppresses the invasive behavior. Furthermore, expression of SMAD4 transcriptionally alters genes associated with the extracellular space, providing evidence that Smad4 regulates the environment surrounding tumors. These results provide new models to study serrated cancer invasion and reveal a role of SMAD4 in manipulating the tumor microenvironment to suppress invasion.

**Abstract:**

Colon cancer is the third most prominent cancer and second leading cause of cancer-related deaths in the United States. Up to 20% of colon cancers follow the serrated tumor pathway driven by mutations in the MAPK pathway. Loss of SMAD4 function occurs in the majority of late-stage colon cancers and is associated with aggressive cancer progression. Therefore, it is important to develop technology to accurately model and better understand the genetic mechanisms behind cancer invasion. Organoids derived from tumors found in the *Smad4^KO^ BRAF^V600E/+^* mouse model present multiple phenotypes characteristic of invasion both in ex vivo and in vivo systems. *Smad4^KO^ BRAF^V600E/+^* tumor organoids can migrate through 3D culture and infiltrate through transwell membranes. This invasive behavior can be suppressed when SMAD4 is re-expressed in the tumor organoids. RNA-Seq analysis reveals that SMAD4 expression in organoids rapidly regulates transcripts associated with extracellular matrix and secreted proteins, suggesting that the mechanisms employed by SMAD4 to inhibit invasion are associated with regulation of extracellular matrix and secretory pathways. These findings indicate new models to study SMAD4 regulation of tumor invasion and an additional layer of complexity in the tumor-suppressive function of the SMAD4/Tgfβ pathway.

## 1. Introduction

Colorectal cancer is one of the most common cancers in the United States. The predominant subtype of CRC is the WNT-driven adenoma. However, approximately 15–30% of patients are diagnosed with the “serrated” subtype of CRC, which is often associated with aggressive progression and high frequency of metastasis [1,2,3,4,5,6,7]. Serrated colon cancers are commonly associated with the BRAF-V600E mutant allele as the truncal mutation [4,5,6,8,9,10,11]. With a poorer prognosis relative to WNT-driven adenomas, serrated CRC remains a significant challenge due to the chance of remaining undetected during screens [12,13] and is relatively understudied when compared to the canonical adenocarcinoma counterparts. More importantly, the genetic progression of serrated CRCs is still being uncovered, and the Smad4/Tgfβ pathway has a substantial role in accelerating serrated cancer progression.

The Smad4/Tgfβ pathway is mutated in 60% of all CRCs and is most often associated with late-stage cancer progression, poor prognosis, and cancer invasion [14,15,16,17]. Almost 30% of serrated cancer patients have an oncogenic driver mutation in SMAD4/TGFβ pathway [10,18,19]. Furthermore, Tgfβ signaling has also been found to be critical in suppression of BRAF-driven oncogenesis in right-sided colon cancers [20], suggesting that Smad4/Tgfβ plays a significant role in serrated colon cancer progression, yet there are still substantial gaps in the current knowledge. The role of SMAD4, specifically during the process of BRAF-mutant tumor invasion, is less understood, partially owing to the limited number of model systems to monitor invasive behavior.

The development of mouse models to study serrated cancers has highlighted differences in the genetic progression of BRAF-driven serrated cancers when compared to the WNT-driven counterparts [9,10,19,21,22,23]. Furthermore, the loss of SMAD4 in oncogenic BRAF-V600E backgrounds results in aggressive invasive behavior in vivo [10,18,19]. The models provide a valuable system to study and assess the critical and complex role of SMAD4 in colon cancer invasion. However, it would be advantageous to develop an ex vivo system to understand the role of SMAD4 in this invasive process. This study reveals that tumor organoids derived from the *Smad4^KO^ BRAF^V600E/+^* mouse model have invasive capabilities ex vivo, providing a controlled model system to study serrated cancer invasion. Additionally, these findings indicate SMAD4 has a direct role in suppressing serrated tumor invasion, and that SMAD4 regulatory targets may shape the tumor extracellular environment, revealing an underappreciated role of SMAD4 in tumor invasion suppression.

## 2. Materials and Methods

### 2.1. Animals

Animal experiments were conducted in accordance with Rutgers University IACUC. Mice 6–8 weeks of age were treated with intraperitoneal injection of tamoxifen (1 mg/20 g) for four consecutive days unless stated otherwise. See Appendix A.

### 2.2. Organoid Culture

Crypt-derived organoids were isolated from duodenum and cultured in Cultrex reduced growth factor matrix R1 (BME-R1) (Trevigen, Gaithersburg, MD, USA) according to established methods [24,25]. Tumor organoids were derived from macroscopic tumor tissue found in *Smad4^KO^; BRAF^V600E/+^; Villin-Cre^ERT2^* mice according to established methods [25]. An average of 100 organoids per biological replicate were seeded in 25 μL of matrix with 1× Crypt Culture Media (CCM) consisting of Basal Crypt Media (BCM): Advanced DMEM/F12 (Gibco, Waltham, MA, USA), 1% Penicillin/Streptomycin (Gibco), 2 mM GlutaMAX (Gibco), 10 mM HEPES (Life Technologies, Carlsbad, CA, USA) supplemented with 50 ng mL^−1^ EGF (R&D, Santa Clara, CA, USA), 100 ng mL^−1^ Noggin (Peprotech, Cranbury, NJ, USA), N-acetyl-l-cysteine 1 μM (Sigma-Aldrich, St. Louis, MO, USA), R-Spondin CM 2.5% (*v*/*v*), 1× N2, 1× B27 (Life Technologies).

### 2.3. Organoid Transduction with pINDUCER-SMAD4

Lentiviral transduction of tumor organoids was performed using established protocol [26] with the following modifications. Lentiviral packaging vectors (pVSVg, Δ8.9) along with mock or pINDUCER-SMAD4 (Baylor) plasmid were incubated with Lipofectamine2000 (ThermoFisher, Waltham, MA, USA) and incubated with HEK293T cells as per manufacturer protocol for 16 h. Cultured medium was refreshed following 16 h incubation and cells were incubated for 72 h, collecting and replacing media after 48 h for first virus collection. Approximately 50 organoids were used per transduction. Organoids were passaged using 1× TrypLE (Gibco) and were incubated with high titer virus in 1× CCM supplemented with 10 mM nicotinamide (Tocris Bioscience, Bristol, UK), 10 µM Chir99021 (Axon MedChem, Reston, VA, USA), 10 µM Y27632 (Tocris Bioscience), and 8 µg/mL polybrene (MilliporeSigma, Darmstadt, Germany) for 4 h. Transduced organoids were seeded in BME-R1 (R&D Systems) and cultured in 1× CCM supplemented with 10 mM nicotinamide, 10 µM Chir99021, and 10 µM Y27632. Successfully transduced organoids were selected for using 2 µg/mL Puromycin (Gibco). To induce expression of SMAD4, organoids were cultured in 1× CCM supplemented with 4 μg/mL doxycycline.

### 2.4. Organoid Imaging

Organoids were passaged after 7 days using 1× TrypLE (Gibco). An average of 100 organoids per biological replicate were seeded and cultured in 1× CCM for 3 days. CCM was then removed, and organoids were treated with fresh 1× CCM supplemented with vehicle control or 4 μg/mL doxycycline to induce SMAD4 expression in pINDUCER-SMAD4 transfected organoids. Images were taken using a light microscope with iPhone7 and iPhoneXR rear-facing cameras.

### 2.5. Organoid Transplantation

Prior to transplantation, immunocompromised mice (NOD.CB17-*Prkdc^scid^*, Charles Rever, Fairfield, NJ, USA) aged 6–8 weeks were given water with dissolved 3.5% Dextran Sodium Sulfate (DSS) (Thermo Scientific) for 5 days. The purpose of this treatment is to induce colon tissue injury and increase the chance for organoids to settle into the tissue of the colon [27]. Mice were allowed to recover for 2 days post-DSS treatment. Weights were recorded on days 0, 3, 5, and 7. The transplantation procedure is as follows. doxycycline -induced tumor organoids were digested in 0.25% trypsin-EDTA (Gibco) in PBS at 37 °C for about 7 min and then were mechanically disassociated using a p200 pipette. Next, the solution was mixed with 10 mL of DMEM (+penicillin/streptomycin +10% FBS) and centrifuged at 1500 rpm. for 5 min. Then, the solution was inspected under a microscope to ensure they remained in clumps. Lastly, they were resuspended to a final concentration of ∼300k cells/50 μL in 5% BME R1 in PBS. Mice were anesthetized using isofluorane (2%) and were kept under until the end of the procedure. All materials were cleaned with Clidox prior to use to reduce the chance of contamination. Using a 20 mL syringe with a plastic tubing addition coated with sterile petroleum jelly, the colon of the mouse was flushed using 20 mL of room temperature 1× Phosphate-Buffered Saline (PBS). Then, 50 μL of organoid BME/PBS solution was injected into the lumen of the colon by a p200 pipette enema for 30 s. The anal verge was sealed with 4 μL of Vetbond Tissue Adhesive (3 M, 1469 SB). Six hours post-transplant, mice were checked to ensure no adverse effects from the transplant.

### 2.6. Tissue Histology and Imaging

Mouse intestines were collected and fixed overnight at 4 °C in a 4% paraformaldehyde solution, washed in PBS three times, fixed overnight at 4 °C in a 30% sucrose solution, and then carefully sectioned into 10 μm OCT sections. Sectioned slides were set out in room temperature for 10 min, washed, submerged in DAPI-mqH_2_O solution for 3 min, washed, and mounted with fluoromount.

For histological analysis, frozen sections were left out to defrost for 10 min and baked at 60 °C for 30 min, re-dehydrated with xylene, decreasing ethanol gradients (100%, 95%, 85%, 70%), and water. They were then submerged in Hematoxylin for 1 min, dipped 3 times in acid alcohol, then submerged in bluing solution for 1 min. Slides were dehydrated through an increasing ethanol gradient and submerged in Eosin for 30s, then sections were mounted.

### 2.7. Western Blot

*Smad4^KO^ BRAF^V600E/+^* tumor organoids were passaged and resuspended in CCM treated with either vehicle or doxycycline (4 μg/mL) for 48 h. Organoids were collected and washed with cold 1× PBS (Gibco) and pelleted to remove BME-R1. Organoids were lysed in RIPA (Life Technologies) supplemented with ProteaseArrest™ (GBiosciences, St Louis, MO, USA). A total of 25 μg of protein was used to perform Western Blot. SMAD4 was detected using anti-SMAD4 (1:3000, Cell Signaling, Danvers, MA, USA) and HRP-tagged secondary (1:5000, Cell Signaling). GAPDH (1:5000, Santa Cruz, Dallas, TX, USA) was used as loading control.

### 2.8. Organoid Immunofluorescence and Imaging

Organoids were passaged after 7 days using 1× TrypLE (Gibco). An average of 100 organoids per biological replicate were seeded and cultured in 1× CCM for 3 days. CCM was then removed and organoids were treated with fresh 1× CCM supplemented with vehicle control or 4 μg/mL Doxycycline to induce SMAD4 expression in pINDUCER-SMAD4 transfected organoids. Organoids were fixed in prewarmed (37 °C) 4% paraformaldehyde (PFA) for 10 min at 37 °C. Organoids were washed 2 times in prewarmed PBS/glycine (100 mM glycine in PBS) for 5 min at room temperature. Organoids were then permeabilized with 0.5% Triton X100 in PBS for 10 min at room temperature. The organoids were blocked using prewarmed IF buffer (0.1% BSA; 0.2% Triton X100; 0.05% Tween-20 in PBS) and 10% serum overnight at room temperature in a humidified chamber with the lid off to prevent drying. Organoids were stained with SMAD4 (1:300, Cell Signaling). The organoids were then washed twice with prewarmed IF buffer for 10 min at room temperature and then stained with secondary Alexa 555 (Invitrogen, Waltham, MA, USA). Organoids were then washed twice with IF buffer for 10 min at room temperature. Organoids were stained with 1:300 DAPI for 15 min at room temperature and then washed twice with prewarmed IF buffer for 5 min at room temperature. IF buffer was removed and Prolong Gold was added dropwise until the organoids were completely covered and then left at room temperature for 20 min to an hour. The Prolong Gold layer was then covered in mineral oil. Organoid images were captured using a Zeiss Axiovert 200 Microscope (Zeiss, Oberkochen, Germany).

### 2.9. RNA-Seq and Analysis

Organoids were passaged and cultured in 1× CCM treated with either vehicle or 4 μg/mL of doxycycline for 48 h prior to collection for RNA-Seq. Organoids were collected and resuspended in Trizol. RNA was extracted using the RNeasy Kit (QIAGEN, Hilden, Germany). A total of 1 μg of RNA was submitted per sample for sequencing (BGI). Resulting raw files were processed using Kallisto [28] and DESeq2 [29] for differential gene expression analysis. Genes were filtered for significantly regulated genes (Log2FC > 1.0 or <−1.0, adj *p*-value < 0.05) for heatmap visualization (http://www.heatmapper.ca/expression (accessed on 20 November 2023)) and DAVID Analysis [30]. GSEA was performed on pre-ranked Signal-2-Noise lists derived as previously published from fpkm tables [10].

### 2.10. ChIP-Seq

Mouse epithelium and CaCo2 SMAD4 ChIP-Seq was performed and data were processed as previously described [31] (GSE112946). SW480 cells were transduced with lentiviral vectors carrying avi-tagged SMAD4 expression constructs (pReceiver-Lv108, GeneCopoeia, Addgene plasmid 29649). SW480 cells stably expressing both avi-tagged SMAD4 and BirA were selected with medium containing 2 μg/mL puromycin and 0.4 mg/mL Hygromycin B, respectively. SW480 ChIP-Seq was performed as previously described with the following modifications. Following cross-linking, SW480 cells were rocked for 20 min at room temperature, after which fresh formaldehyde was added to make a concentration of 1.22% formaldehyde. Sonication was performed with a final concentration of 0.2% SDS in the sonicates. Diluted sonicates were then incubated with pre-blocked (0.5% BSA/PBS) Streptavidin beads (Invitrogen) overnight at 4 °C. The beads were serially washed in low-salt (0.1% SDS, 0.1% Sodium Deoxycholate, 1% Triton X-100, 150 mM NaCl, 1 mM EDTA, and 20 mM HEPES, pH 8.0), high-salt (0.1% SDS, 0.1% Sodium Deoxycholate, 1% Triton X-100, 500 mM NaCl, 1 mM EDTA, and 20 mM HEPES, pH 8.0), LiCl buffer (250 mM LiCl, 0.5% Sodium Deoxycholate, 0.5% NP-40, 1 mM EDTA, and 20 mM HEPES, pH 8.0), and a final wash buffer (1 mM EDTA and 20 mM HEPES, pH 8.0). A total of 5 ng each of purified ChIP or input DNA was used to prepare ChIP-seq libraries. Fragment size was selected with Pippin Prep and sequenced on Illumina HiSeq 2500 (Illumina, San Diego, CA, USA).

The quality of sequenced reads was assessed with FastQC (v.0.11.3) and bowtie2 [32] (v.2.2.6) was used to align the sequences to mouse (mm9) or human (hg19) genomes. Deeptools bamCoverage (v.2.4.2) was used to generate bigwig files for visualization using Integrative Genomics Viewer [33] (IGV 2.4.13). MACS (1.4.1) [34] was used for peak calling from aligned reads, with peaks being filtered at a *p*-value of 10^−5^. Genomic Regions Enrichment of Annotations Tool (GREAT v4) [35] was used to identify the two nearest genes within 50 kb of SMAD4-bound sites. SMAD4-bound genes and significantly up- and downregulated genes from RNA-Seq were plotted in GeneVenn. SW480 and CaCo2 SMAD4-bound genes were converted to mm9 gene annotations using SynGO ID convert tool [36].

### 2.11. Transwell Invasion Assay

Organoid invasion assay was performed as previously described with the following modifications [37]. *Smad4^KO^ BRAF^V600E/+^* tumor organoids were passaged and resuspended in CCM. A total of ~50 organoids were transferred into individual transwell inserts of the 24-well plate and incubated at 37 °C. After 21 days, transwell inserts were removed from the plate. Organoids were fed with CCM every other day. Images were taken under an inverted microscope with Samsung Galaxy S10 and iPhone following media exchange. After 7 days post-transwell removal, organoid colonies that were established were counted.

### 2.12. Statistical Analyses

Graphs were made in GraphPad Prism (v8+), and SEM was plotted. Student’s *t*-test was performed for paired comparisons while ANOVA was used for multiple comparisons within GraphPad Prism (v8+).

## 3. Results

### 3.1. Smad4^KO^ BRAF^V600E/+^ Tumor Organoids Exhibit Invasive-like Behavior

Mice induced to lose SMAD4 and simultaneously activate *BRAF^V600E/+^* in the gastrointestinal tract develop serrated tumors, which can efficiently advance to invasive carcinomas [10,19]. To better understand the progression of the tumor epithelium to invasive behavior, tumor organoids were derived from *Smad4^KO^ BRAF^V600E/+^* mice and plated in a 3D culture environment (BME R1) [23]. Unlike the budded crypt structures seen in wildtype and non-tumor organoids, tumor organoids typically maintain a spherical morphology, indicative of a “stem-like de-differentiated” state of the cells [19,24,25,38]. Interestingly, *Smad4^KO^ BRAF^V600E/+^* tumor organoids have a shift in morphology where the organoids are capable of expansion and develop “projections” within BME R1, which protrude through the extracellular matrix and often fuse with neighboring organoids (Figure 1A). These organoids further penetrate towards the bottom of wells and migrate on the base of the culture plates. Fluorescence microscopy of *Smad4^KO^ BRAF^V600E/+^* tumor organoids expressing the *Rosa-CAG-LSL-ZsGreen1-WPRE* (RosaGFP) allele [39] reveal that the projections are cellular and originate from the organoids themselves (Figure 1B). This phenotype is similar to morphological changes that are found in other invasive cancer organoids [40,41,42]. Notably, only the tumor organoids could develop projections and connections with neighboring organoids, revealing that the invasive behavior is unique to the tumors (Figure 1C, * *p*-value < 0.05, Student’s *t*-test).

To confirm the invasive potential of the *Smad4^KO^ BRAF^V600E/+^* tumor organoids shown in vitro, *Smad4^KO^ BRAF^V600E/+^* RosaGFP tumor organoids were transplanted into NOD.CB17-*Prkdc^scid^* mice (*n* = 12) [27]. A total of 2 months after transplant, GFP expression was observed in seven of twelve mice, which developed macroscopic dysplasias. Five of seven mice harbored invasive lesions with GFP-positive tumor organoid cells penetrating through the muscle layer—consistent with invasive lesions seen previously in *Smad4^KO^ BRAF^V600E/+^* mice (Figure 1D, Appendix A) [10,19]. These findings suggest that the tumor organoid formation of projections in vitro may correspond to invasive behavior in vivo.

Oncogenic mutations activating the WNT signaling pathway were a universal feature of serrated tumors in the model *Smad4^KO^ BRAF^V600E/+^*, indicating that the combination of BRAF, SMAD4, and WNT mutations is critical for the progression of serrated colon cancers [9,18,19,21]. This raises the possibility that the invasive behavior is dependent on oncogenic WNT signaling, as opposed to SMAD4 loss. To assess this, wildtype, *BRAF^V600E/+^ β-catenin^gof^,* and *Smad4^KO^ BRAF^V600E/+^ β-catenin^gof^* Villin-Cre mice were harvested for organoids, cultured for 7 days, and imaged for capability to form projections (Figure 1E). Wildtype organoids were unable to develop projections as expected. While some *BRAF^V600E/+^ β-catenin^gof^* organoids did show the capability to develop projections, the *Smad4^KO^ BRAF^V600E/+^ β-catenin^gof^* organoid lines extended significantly more projections that often connected to neighboring organoids (Figure 1F). These findings confirm that while the combination of oncogenic SMAD4, BRAF, and WNT mutations is critical for serrated cancer progression, the loss of SMAD4 is the critical driver for organoid invasive capability.

### 3.2. SMAD4 Suppresses Invasive Behavior in Tumor Organoids

SMAD4 has long been associated as a late-stage colon cancer mutation that results in poor prognosis and metastasis [43,44,45,46]. To test whether SMAD4 can directly suppress invasive behavior, *Smad4^KO^ BRAF^V600E/+^* tumor organoids were engineered to harbor a pINDUCER-*SMAD4* construct [47]. Independent pINDUCER-SMAD4 *Smad4^KO^ BRAF^V600/+^* tumor organoid lines were selected using puromycin (2 μg/mL), passaged, and treated for 48 h with doxycycline (dox, 4 μg/mL) to induce SMAD4 expression. Western blotting and immunofluorescence confirmed SMAD4 expression upon doxycycline treatment, confirming the integration of the construct (Figure 2A,B and Appendix A). To assess whether the expression of SMAD4 in tumor organoids resulted in a suppression of the invasive behavior, pINDUCER-SMAD4 tumor organoids were passaged and either treated with vehicle or dox. Vehicle-treated organoids developed projections and were able to migrate through the matrix as previously seen, while pINDUCER-SMAD4 tumor organoids expressing SMAD4 showed that many organoids retained spherical morphology and did not appear to develop projections (Figure 2C). Quantification revealed that SMAD4 expression significantly reduced the number of projections made by the organoids (Figure 2D, * *p*-value < 0.05, Student’s *t*-test). With the inability to form projections, EdU incorporation and imaging was performed to determine if cell proliferation was impacted by the expression of SMAD4. Expectedly, tumor organoids have actively proliferating cells decorating the spheroids. However, expression of SMAD4 in tumor organoids has no significant difference in proliferating cells (Figure 2E,F). These results reveal that SMAD4 expression can directly impact the invasive behavior of serrated tumor organoids yet does not appear to have a short-term effect on cell proliferation.

### 3.3. SMAD4 Regulates Extracellular Environment Genes

With the impact of SMAD4 in tumor organoid migratory and invasive capability, it was of interest to understand the transcriptional mechanism of how SMAD4 suppresses this invasive behavior. pINDUCER-SMAD4 *Smad4^KO^ BRAF^V600E/+^* tumor organoids were treated with either vehicle (*n* = 4) or dox (*n* = 2) to induce expression of SMAD4 for 48 h and then collected for RNA-Seq. GSEA analysis confirmed that expression of SMAD4 resulted in a significant upregulation of hallmark TGFβ-pathway targets and there was also a suppression of MYC targets (Figure 3A, K-S Test), consistent with the antagonistic impact of SMAD4 expression on WNT signaling [38,48,49,50,51]. DESeq2 analysis revealed that 278 genes were significantly regulated by SMAD4. Of those, 163 were significantly upregulated, whereas 115 were significantly downregulated (Figure 3B, Appendix A). Significantly upregulated genes were most closely associated with secreted proteins and the extracellular matrix (Figure 3C). Conversely, other secreted and extracellular matrix-associated genes were downregulated (Figure 3D), suggesting that SMAD4 may regulate a complex interaction with the extracellular environment. Tumor organoids were also found to have an upregulation in integrins as compared to wildtype counterparts, further suggesting that extracellular interactions are being impacted (Appendix A). To determine if SMAD4 is a direct regulator of these genes, SMAD4 ChIP-Seq of mouse epithelium and human colon cancer cell lines was overlayed with the significantly regulated genes (Appendix A). The subset of genes that were associated with SMAD4 binding were also found to be enriched for extracellular pathway-associated genes in both upregulated (Appendix A) and downregulated gene sets (Appendix A). These results suggest that SMAD4 regulates how the cells interact with the extracellular environment.

### 3.4. SMAD4 Loss Enables Organoid Survival Independent of Extracellular Matrix

The capability of *Smad4^KO^ BRAF^V600E/+^* tumor organoids to penetrate and migrate through BME R1 and adhere to the bottoms of the culture wells suggests that tumor organoids are capable of growth in a 2D monolayer similar to cancer cell lines and may survive independent of 3D culture conditions. Wildtype, *BRAF^V600E/+^ β-catenin^gof^*, *Smad4^KO^ BRAF^V600E/+^* non-tumor, and *Smad4^KO^ BRAF^V600E/+^* tumor organoids were cultured in BME R1, then passaged into tissue culture-treated dishes without BME R1. By day 5 after plating, the majority of wildtype and *BRAF^V600E/+^ β-catenin^gof^* organoids were unable to survive at the base of the culture wells. Conversely, approximately 50% of *Smad4^KO^ BRAF^V600E/+^* non-tumor organoids and almost all tumor organoids survived on tissue culture plastic (Figure 4A). While some of the non-tumor *Smad4^KO^ BRAF^V600E/+^* organoids formed a 2D monolayer, the majority of tumor organoids were capable of this feat (Figure 4B, Appendix A). This suggests that SMAD4 loss is critical for the ability of organoids to sustain themselves independent of the extracellular matrix typical of 3D organoid cultures.

The ability of *Smad4^KO^ BRAF^V600E/+^* tumor organoids to grow through BME R1 and onto plastic is reminiscent of invasive growth similarly found in 2D invasive cancer cell lines [52,53]. Thus, we employed a transwell assay to further test whether SMAD4 impacts invasive capability [54,55]. Tumor organoids were passaged from BME R1 and plated on a transwell assay plate without BME R1. Organoids were allowed to settle onto the membrane (3 days), and then incubated for 21 days. Transwell inserts were then removed and the cells that were able to extrude through the membrane were allowed to settle and establish colonies for 7 days following removal of transwell (Figure 4C, Appendix A). This assay revealed that tumor organoids were capable of penetrating through the transwell membrane and ultimately formed colonies on the bottom of the plate, further indicating their invasive potential (Figure 4D).

To determine whether SMAD4 impacts this invasive behavior through transwell membranes, pINDUCER-SMAD4 tumor organoids were seeded into the transwell with or without dox to induce expression of SMAD4 (Figure 5A). After a total of 28 days, organoids were assessed for capability to penetrate the membrane and sustain themselves at the bottom of the plate (Figure 5B). Vehicle-treated tumor organoids saw on average 40 colonies formed on the bottom of the plastic. In contrast, tumor organoids re-expressing SMAD4 only saw 10 colonies, showing a lower efficiency of penetration through transwell membranes (Figure 5C). These findings suggest that SMAD4 is critical in suppressing the invasive ability of the tumor organoids.

### 3.5. SMAD4 Differentially Regulates Genes Dependent on Environment

Since tumor organoids grew independently of extracellular matrix, it was important to determine how the expression of SMAD4 would alter the transcriptome in this new extracellular environment. pINDUCER-SMAD4 tumor organoids were passaged and seeded onto plastic and then supplemented with either vehicle or 4 μg/mL dox, and then submitted for bulk RNA-Seq. SMAD4 expression on organoids growing in the absence of BME induced a regulatory shift in 997 genes, with 641 genes upregulated and 356 genes downregulated (Figure 6A, Appendix A). Similar to organoids grown in 3D culture (Figure 3), DAVID analysis indicated an upregulation in pathways associated with secretory proteins and extracellular regions upon re-expression of SMAD4 in tumor organoids cultured on plastic (Figure 6B). In contrast, however, organoids growing on plastic exhibited transcriptome shifts consistent with cell cycle inhibition upon re-expression of SMAD4, a characteristic not significantly observed in organoids cultured in BME R1 (Figure 6). To assess the impact of SMAD4 expression on 2D organoid proliferation, vehicle- and dox-treated pINDUCER-SMAD4 organoids grown on plastic for 5 days were treated with EdU for 4 h. In vehicle-treated wells, the tumor organoids showed robust proliferation and appeared to have a directionality bias towards neighboring organoids. However, dox-treated wells revealed a significant decrease in EdU staining with organoids and showed loss of directionality (Figure 6D–F). GSEA analysis confirmed that TGFβ signaling was significantly upregulated upon expression of SMAD4. Notably, hallmark cell cycle and both G1 and G2/M cell cycle checkpoint pathways [56] were significantly downregulated in the RNA-Seq data (Figure 6F). Comparatively, GSEA of the pINDUCER-SMAD4 organoids in 3D culture reveals that only the G2/M checkpoint pathway was significantly downregulated (Appendix A), and it was only after SMAD4 was being expressed for 5 consecutive days that cell cycle genes were suppressed in BME R1 cultured organoids (Appendix A). This suggests that SMAD4 expression can suppress the cell cycle but has a pronounced effect when cultured in the absence of BME R1 and requires prolonged expression of SMAD4 to induce cell cycle regulation. These findings confirm that the expression of SMAD4 can suppress cell proliferation [38,49,57] in the absence of the 3D culture and possibly exacerbates invasion inhibition.

While the expression of SMAD4 is shown to suppress the invasive behavior and transcriptionally regulate extracellular region genes and cell cycle, it is important to discern the direct role of SMAD4 on these genes. Smad4 ChIP-Seq was performed in whole epithelium in mouse, and in CaCo2 and SW480 cancer cell lines [31]. SMAD4-enriched binding sites were then mapped to genes with transcriptional start sites within 50 kb. Genes with SMAD4 binding sites were compared to the significantly up- and downregulated genes from the pINDUCER-SMAD4 tumor organoid RNA-Seq to identify potential direct target genes of SMAD4 activity (Figure 7A, Appendix A). Of the ~14,000 Smad4-enriched binding sites in the whole-epithelium ChIP-Seq, 411 genes that were upregulated upon re-expression of SMAD4 were also bound by SMAD4, while 189 genes that were downregulated were bound by SMAD4. In human CRC cell lines, 457 genes that were upregulated were bound by SMAD4, and 219 genes were downregulated upon re-expression of SMAD4. DAVID analysis of genes upregulated and bound by SMAD4 in the whole epithelium showed that they were predominantly associated with membrane and lysozyme proteins (Figure 7B). While similar pathways were found in the human cell line ChIP-Seq, the predominant association was with secretory pathway genes (N-linked asparagine) (Figure 7B). Conversely, SMAD4 bound genes that were downregulated in the RNA-Seq data were associated with the cell cycle (Figure 7C). This suggests that SMAD4 has a role in suppressing the cell cycle by directly binding to these genomic regions. RNA-Seq data confirm that both G1/S Phase- and G2/M Phase-associated genes are suppressed upon expression of SMAD4 within the tumor organoids (Figure 7D). Furthermore, IGV traces of these cell cycle regulatory genes reveal that SMAD4 is enriched at the promoter regions of these cell cycle genes (Figure 7E), confirming that SMAD4 has a direct role in transcriptionally regulating cell cycle genes.

## 4. Discussion

### 4.1. Loss of SMAD4 Promotes BRAF-Driven Serrated Cancer Invasion

The canonical WNT-driven adenocarcinoma genetic requirements for invasion and metastasis consider SMAD4 a late-stage driver mutation for invasion and metastatic cancers [14,45,58,59,60]. SMAD4 expression and Tgfβ inhibitors in CRC cell lines limit cancer cell migration, thus suppressing metastatic progression [50,61,62]. Recent studies also reveal that the loss of SMAD4 promotes metastatic capability in AKP organoids when orthotopically transplanted into mice [16,17,63]. While the invasive and metastatic process is studied in WNT-driven adenomas, the BRAF-driven serrated cancer invasion process is still relatively understudied.

Tumor organoids derived from *Smad4^KO^ BRAF^V600E/+^* mice develop a unique phenotype to form cellular projections that penetrate and migrate through BME-R1—a phenotype that is also seen in other organoid systems [64,65,66,67]. Orthotopic transplantation of these tumor organoids shows that the organoids are also capable of invasion in vivo. This finding confirms that tumor organoids can emulate the invasive properties of the primary tumor from the original mouse model [10,18,19], providing a valuable model system to assess invasive cancer behavior in a controlled environment. With the capability of tumor organoids to exhibit invasive behavior, it adds to the depth of possibilities for organoid models [68,69,70].

While BRAF-V600E is the primary oncogenic driver of serrated CRC, the activation of WNT is also a critical factor in serrated cancer initiation and progression. WNT and SMAD4/BMP signaling have an antagonistic role in intestinal homeostasis, and WNT is a critical oncogenic driver predominantly found in CRCs [49,71,72]. The loss of SMAD4 results in strong selective pressure for gain-of-function WNT mutations [9,10,18,19,21,22,23], and has been found to be a key driver in transcriptional reprogramming to drive long-term WNT independence in CRCs [73]. This may indicate that the oncogenic WNT activation in these tumor organoids is the critical factor in promoting invasion. *Smad4^KO^ BRAF^V600E/+^ β-catenin^gof^* mice have previously been shown to have aggressive progression of serrated dysplasias with invasive capability while *BRAF^V600E/+^ β-catenin^gof^* mice did not [19]. Similarly, this study reveals *Smad4^KO^ BRAF^V600E/+^ β-catenin^gof^* organoids have the same invasive behaviors seen in *Smad4^KO^ BRAF^V600E/+^* tumor organoids, whereas *BRAF^V600E/+^ β-catenin^gof^* organoids did not, suggesting that the loss of SMAD4 was critical for the invasive behavior.

### 4.2. SMAD4 Directly Suppresses Organoid Invasive Capabilities In Vitro

The growing prevalence of organoids has promoted the study of three-dimensional migration and invasive phenotypes [54]. However, the capability of tumor organoids to grow independently of 3D culture also allows for other invasive assays to be considered. Here, we use transwell invasion assays for *Smad4^KO^ BRAF^V600E/+^* tumor organoids which also have the capability to grow in 2D. Surprisingly, the transwell assay reveals the capability of tumor organoid cells to invade through the membrane similar to cancer cells. These results along with other recent BRAF-V600E-focused studies suggest that BRAF-driven serrated CRCs are already “primed” for invasion upon tumor initiation, and that SMAD4 loss is crucial to promote invasion [18,19,22].

With the impact of SMAD4/Tgfβ pathway on cancer invasion, it raises the question of how SMAD4 directly impacts cancer invasion, particularly in colon cancers. Using the pINDUCER-SMAD4 construct in our organoid system, it can be directly tested whether the aggressive invasive capability of *Smad4^KO^ BRAF^V600E/+^* tumors is dependent on SMAD4 loss. Interestingly, SMAD4 re-expression within the tumor organoids results in a significant reduction in invasive capability, indicative of a direct role of SMAD4 in suppression of serrated cancer invasion.

### 4.3. SMAD4-Dependent Suppression of Invasion Differs Based on Extracellular Environment

The tumor-suppressive function of SMAD4 has often been associated with the suppression of WNT and proliferation, particularly in the intestinal tract [38,48,50,74,75]. Our findings support this canonical notion that SMAD4 expression in the tumor organoids resulted in the suppression of WNT and proliferation, and ChIP-Seq of SMAD4 reveals binding sites at prominent cell cycle genes [19,38,48,49,50,76]. However, EdU staining of our pINDUCER-SMAD4 tumor organoids revealed no substantial difference in proliferation while organoids were in BME R1. This suggests that the tumor-suppressive role of SMAD4 may be more complex than just simple antagonism of proliferation. As such, it would be interesting to assess whether SMAD4 is directly involved with the regulation of cell cycle independent of WNT and if it is conditional on the surrounding environment.

If suppression of proliferation is not the immediate impact of SMAD4 expression, then how does SMAD4 directly impede cancer invasion? RNA-Seq analysis reveals that SMAD4 expression in the invasive tumor organoids resulted in an immediate transcriptional shift in genes associated with the extracellular environment—both in 3D and 2D cultures. ChIP-Seq coincides with these findings as SMAD4 binding sites also associate with significantly regulated genes that regulate ECM and secretory pathways. Previous studies have shown that cancer cells respond differently to different extracellular environments, which may promote invasiveness [77,78,79,80,81,82]. Organoids have also been shown to grow differently in Collagen I-based matrigels as opposed to Collagen IV-based gels (including BME-R1) [77,78,83]. Thus, it is exciting to consider that SMAD4 expression would respond differently based on the environment, and how this influences cancer progression. For instance, SMAD4 has a well-documented role in altering the tumor microenvironment to promote metastasis [79,84]. It has also been shown in multiple cell lines that Smad4/Tgfβ/BMP expression can shift and regulate ECM stiffness to enhance cell invasion and motility [14,81,85]. Finally, loss of SMAD4 in metastatic tumor organoids has increased secretion of DKK3, which inhibits NK cell activity, providing evidence of SMAD4 loss influencing tumor immune response [86]. Therefore, a SMAD4-dependent response and adaptation to multiple environments may be critical in suppression of invasion.

## 5. Conclusions

This study reveals that SMAD4 is a critical tumor suppressor for BRAF-driven serrated tumors. Organoids derived from aggressive *Smad4^KO^ BRAF^V600E/+^* tumors have invasive capabilities both in vivo and in vitro, providing a powerful tool to study the process of cancer invasion in a controlled and manipulatable system. The re-expression of SMAD4 in invasive organoids shows a direct impact of SMAD4 in suppressing invasion, though the proliferation in organoids remained intact, suggesting a tumor-suppressive role of SMAD4 beyond just WNT-antagonism.

In support of this notion, re-expression of SMAD4 in organoids transcriptionally shifts genes associated with extracellular space and secretory pathways. This study suggests that the role of SMAD4 as a tumor suppressor is more complex than currently appreciated and may have implications in the understanding of cancer invasion. Already, new PDO studies have suggested that new therapies and targets for late-stage cancer treatments must consider the surrounding environment [79,84,87,88]. Thus, it would be exciting to further delve into the conditions for SMAD4-dependent gene regulation in different tumor microenvironments, and the continued use of organoid modeling provides a powerful foundational tool in these future studies.

## Figures and Tables

**Figure 1 cancers-15-05820-f001:**
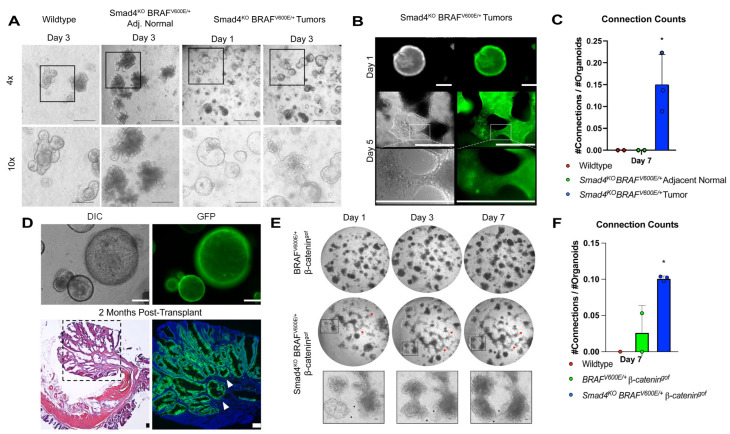
Smad4^KO^ BRAF^V600E^ tumor organoids have invasive capability in 3D environment. (**A**) Representative images of organoids cultured in BME R1. Adjacent normal and tumor organoids were derived from *Smad4^KO^ BRAF^V600E/+^ Villin^Cre-ERT2^* mice 3–6 months post-tamoxifen injection. Tumor organoids have capability to migrate through Matrigel and develop projections which connect neighboring organoids. Scale bars = 0.5 mm. (**B**) Tumor organoids with RosaGFP expression reveal that projections are cellular. Scale bars = 0.5 mm (**C**) Capability to develop projections is unique to tumor organoids (* = *p*-value < 0.05, Student’s *t*-test). (**D**) *Smad4^KO^ BRAF^V600E/+^ Villin^Cre-ERT2^* Rosa-GFP tumor organoids transplanted into NOD.CB17-*Prkdc^scid^* mice colon for 2 months revealed invasive capability (white arrows). Scale bars = 0.5 mm (**E**) *Smad4^KO^ BRAF^V6000E/+^ β-catenin^gof^ Villin^Cre-ERT2^* organoids developed similar invasive behavior as *Smad4^KO^ BRAF^V600E/+^* tumor organoids (red arrows). (**F**) Capability to form projections was highest in *Smad4^KO^ BRAF^V600E^ Ctnnb1^Exon3^ Villin^Cre-ERT2^* organoids (* = *p*-value < 0.05, Student’s *t*-test).

**Figure 2 cancers-15-05820-f002:**
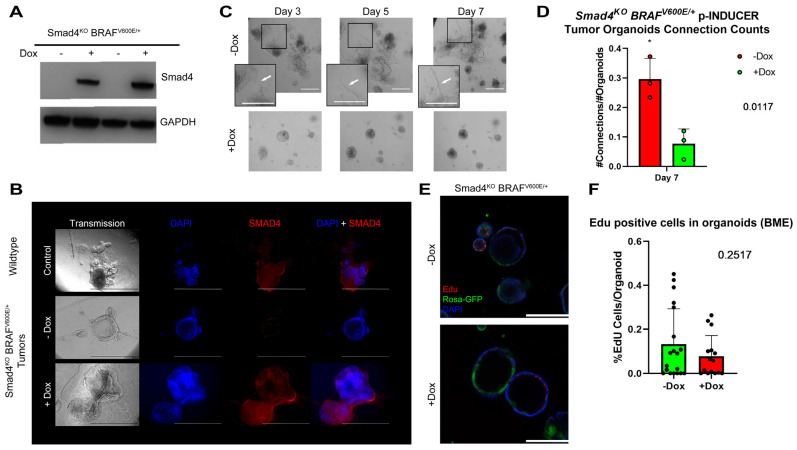
SMAD4 expression in tumor organoids suppresses invasive capability. (**A**) Western Blot of doxycycline induced expression of *SMAD4* in pINDUCER-SMAD4 *Smad4^KO^ BRAF^V600E/+^* tumor organoids. (**B**) Immunofluorescence images of wildtype organoids (Control) and untreated (−Dox) and treated (+Dox) pINDUCER-SMAD4 *Smad4^KO^ BRAF^V600E/+^* tumor organoids. (**C**) Representative images displaying growing cellular connections (white arrows) between tumor organoids in either untreated or treated conditions. Scale bar = 0.5 mm. (**D**) *Smad4^KO^ BRAF^V600E/+^* tumor organoids form significantly more connections than *Smad4^KO^ BRAF^V600E/+^* organoids with induced expression of *Smad4* via doxycycline. (* = *p*-value = 0.0.017, Student’s *t*-test). 10× transmission and florescence images representative of 3 biological replicates. (**E**) EdU stain of tumor organoids treated with either vehicle or doxycycline. 10× images representative of 3 biological replicates (Scale bar = 0.5 mm). (**F**) Quantification of EdU-positive cells in pINDUCER-SMAD4 *Smad4^KO^ BRAF^V600E/+^* organoids.

**Figure 3 cancers-15-05820-f003:**
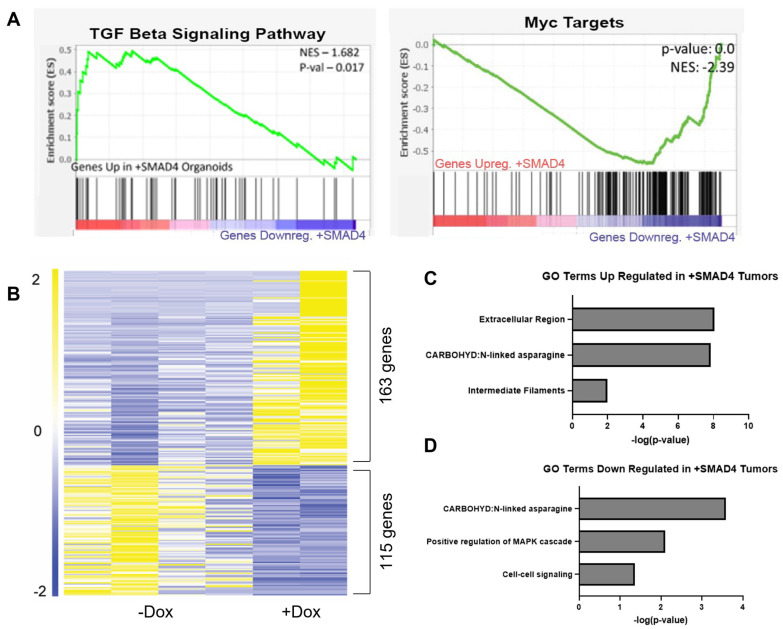
SMAD4 expression alters ECM and secretory pathway genes. (**A**) GSEA analysis shows Tgfβ signaling genes are significantly upregulated in SMAD4-expressing tumor organoids. GSEA analysis shows MYC targets are significantly downregulated in SMAD4+ organoids grown in BME. (**B**) Heatmap of significantly regulated (log2FC < −1 and >1, adj *p*-value < 0.05) pINDUCER-SMAD4 *Smad4^KO^ BRAF^V600E/+^* organoids untreated (−Dox) (*n* = 4) and treated (+Dox) with 4 μg/mL Doxycycline treatment (*n* = 2). (**C**) GO Terms show significant upregulation in gene sets related to extracellular region, intermediate filaments, and n-linked asparagine in tumor organoids grown in BME with expression of SMAD4 induced via doxycycline. (**D**) GO Terms show significant downregulation in gene sets related to positive regulation of MAPK cascade, cell-to-cell signaling, and n-linked asparagine in tumor organoids grown in BME with expression of SMAD4 induced via doxycycline.

**Figure 4 cancers-15-05820-f004:**
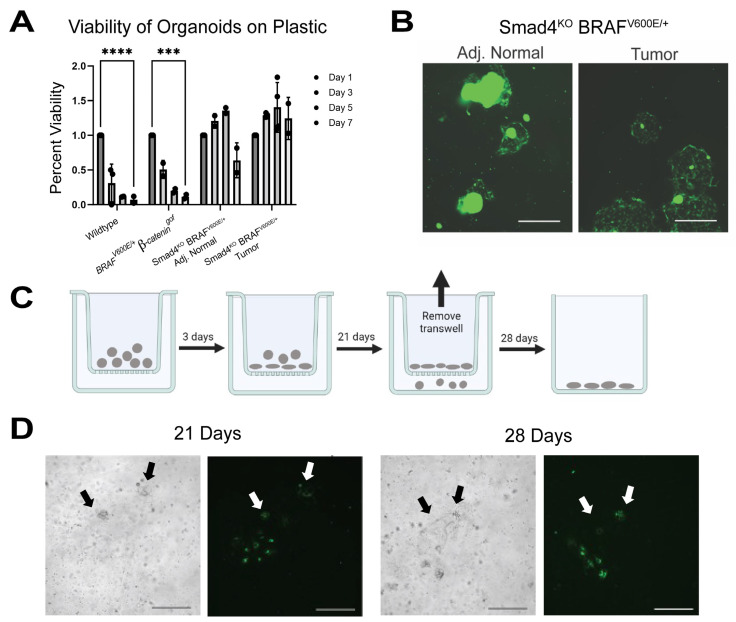
*Smad4^KO^ BRAF^V600E/+^* tumor organoids have invasive behavior in 2D culture. (**A**) Viability of organoids plated without BME (*** = adj *p*-value = 0.002; **** = adj *p*-value < 0.0001, ANOVA). (**B**) *Smad4^KO^ BRAF^V600E/+^* adjacent normal and tumor organoids are capable of surviving on plastic and formed a 2D monolayer. Scale bar = 1 mm. (**C**) Diagram of transwell assay for invasive potential. (**D**) *Smad4^KO^ BRAF^V600E/+^* RosaGFP tumor organoids were capable of migrating through transwell after 3 weeks and forming 2D colonies (Arrows). Images representative of 3 biological replicates. Scale bar = 0.5 mm.

**Figure 5 cancers-15-05820-f005:**
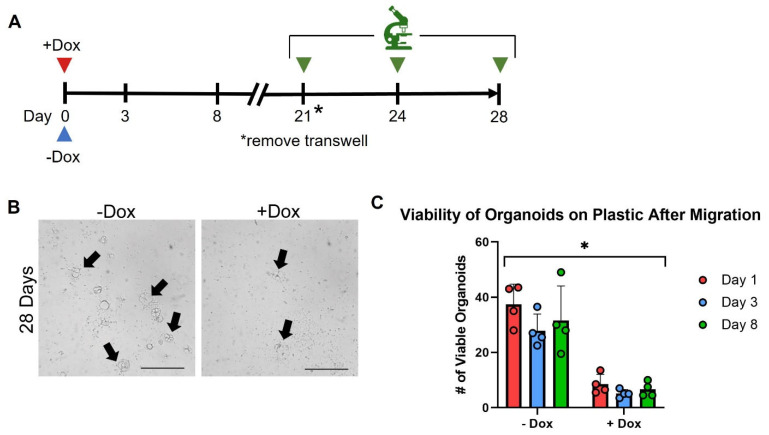
SMAD4 suppresses organoid invasion in 2D culture. (**A**) Timeline of transwell assay and treatment with doxycycline to induce SMAD4. (**B**) Images of organoid colonies on plastic (arrows) representative of 3 biological replicates. Scale bar = 1 mm. (**C**) Tumor organoids expressing SMAD4 prior to transwell removal were unable to migrate. (* = *p*-value < 0.05, ANOVA).

**Figure 6 cancers-15-05820-f006:**
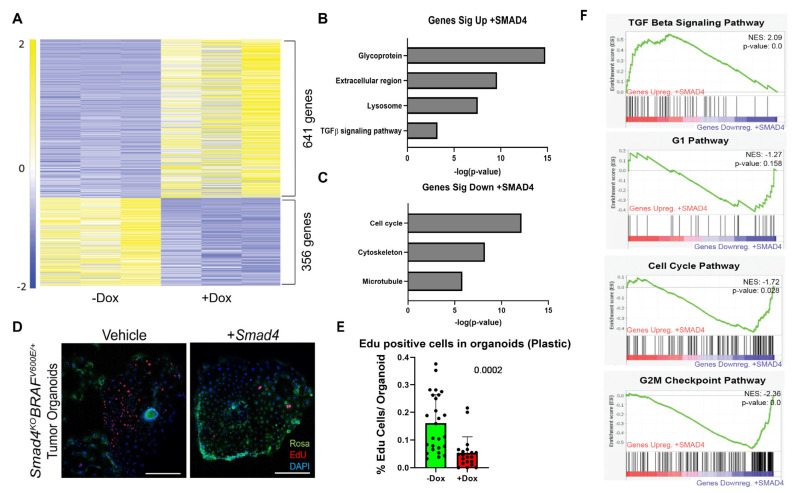
SMAD4 suppresses cell cycle and proliferation in 2D culture. (**A**) Heatmap of Dox-treated orgnaoids in 2D monolayer. (**B**,**C**) DAVID analysis of genes upregulated (**B**) and downregulated (**C**) in SMAD4-expressing tumor organoids grown in the absence of BME R1. (**D**) EdU stain of tumor organoids in 2D cultures, images representative of 3 biological replicates. Scale bar = 1 mm. (**E**) Quantificaiton of EdU-positive cells in organoids (*p*-value reported, Student’s *t*-test). (**F**) GSEA analysis reveals cell cycle is downregulated upon expression of SMAD4.

**Figure 7 cancers-15-05820-f007:**
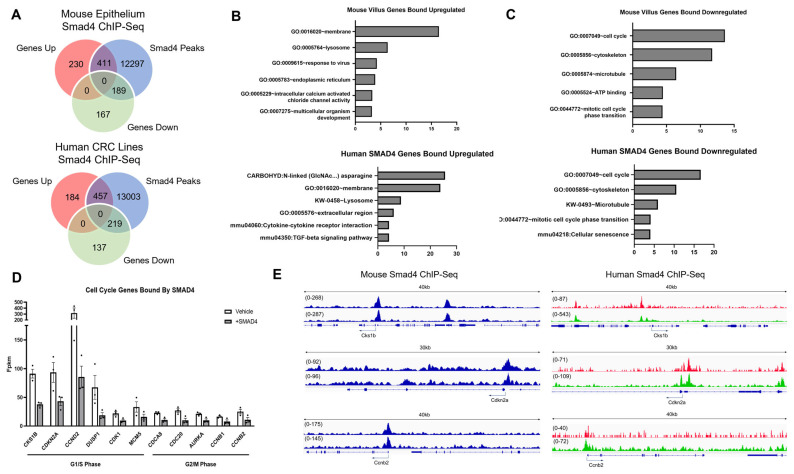
Smad4 directly binds to cell cycle genes downregulated upon re-expression of SMAD4. (**A**) Smad4 binding sites from mouse whole epithelium (*n* = 2), or CaCo2 (*n* = 1) and SW480 (*n* = 1) human CRC cells. ChIP-Seq data are compared with significantly regulated genes from RNA-Seq of tumor organoids +/− SMAD4 (from Figure 6, log2FC > 1 or <−1, adj *p*-value < 0.05). (**B**) DAVID analysis of genes directly bound by SMAD4 and upregulated and (**C**) downregulated. (**D**) RNA-Seq of cell cycle genes bound by SMAD4 and significantly downregulated (adj *p*-value < 0.05). (**E**) IGV of cell cycle genes in SMAD4 ChIP-Seq from mouse epithelium (*n* = 2, blue), SW480 (red), and CaCo2 (green).

## Data Availability

RNA-Seq and ChIP-Seq files are deposited in GEO (GSE232137, GSE232136, GSE112946).

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
