# Peer review of "In Vitro Organoid-Based Assays Reveal SMAD4 Tumor-Suppressive Mechanisms for Serrated Colorectal Cancer Invasion"

_cancers, 2023, doi:10.3390/cancers15245820_

Round 1

Reviewer 1 Report

Comments and Suggestions for Authors

Dear authors,

thank you for this valuable piece of work. Should the other reviewer(s) recommend edits to your work, please have a look at: https://www.mdpi.com/2072-6694/14/13/3252

This may spice up your discussion.

Kind regards

Author Response

Dear authors,

thank you for this valuable piece of work. Should the other reviewer(s) recommend edits to your work, please have a look at: https://www.mdpi.com/2072-6694/14/13/3252

This may spice up your discussion.

Kind regards

We thank the reviewer for their kind remarks regarding our paper. We also appreciate the reviewer for highlighting a recent paper looking at drug sensitivies in PDOs with SMAD4-KO and BRAF mutations. This is very exciting work that supports our use of organoids to study BRAF-driven serrated cancers, and has been added and described accordingly.

Reviewer 2 Report

Comments and Suggestions for Authors

The article is well crafted and comprehensive; however, I have some suggestions for presentation improvement:

The title is clear and concise.

The abstract is well-written and effectively summarizes the study.

The last paragraph of the introduction lacks a clear study goal; consider refining and removing suggestions or transferring them to the conclusion.

The statistical analysis appears questionable, and using a t-test for comparing three groups may not be appropriate. Please provide a complete list of all tests used in the study.

The results section is clear and informative.

Consider subdividing the discussion and incorporating more subtitles, starting with an initial one for clarity.

It is recommended to present a distinct and separate conclusion section.

Update references older than 2017 and ensure all citations conform to the journal's formatting instructions.

Overall, addressing these suggestions will enhance the presentation and strengthen the overall impact of your article.

Reviewer 3 Report

Comments and Suggestions for Authors

Dear Dr. Tong

thank you for that nice research article presenting an in vitro model that reveals SMAD4 dependent invasive behaviour in tumor organoids in a BRAF-V600E background. 

In general your manuscript is well written and results are presented clearly. I just have some minor issues: 

Reference: 
- line 228: Hira et al 2020 
is not listed in your references
- please include recenlty published articel: https://doi.org/10.1016/j.trecan.2023.09.011

Introduction: 
- line 33- 35:  "Critically,the patient demographic is getting younger, to the point where the CDC has recommended earlier screening ages"
relevance of this information in your research context?

- line 40/41: "serrated CRC remains a significant challenge due to the chance of remaining undetected during screens"
why does serrated CRC remain offten undetected? (just a matter of interest) 

-line 45/47: redundant. can you combine both information into 1 sentence?

Results: 
- It's hard to understand that in some paragraphs oncogenic activation of WNT signaling pathwa is taken into account and in the follwoing section it's not. For example: 3.1 shows, that the combination of SMAD4, BRAF and WNT mutations can lead to progression to invasive behaviour but in the next section. SMAD induction is analysed in the background of BRAF alone. While in section 3.4 b-catenin is taken into account again, but not in SMADko organoids. Can you please explain/ introduce your sections in more detail.

- EdU incorporation: It's hard to draw any conclusions from images provided in figure 2E/ 6D. Can you please quantify the proliferation of your organoid culture?

- section 3.4: can you please specify why organoids migrating in BME should survive in a 2D monolayer and survive independent of 3D culture conditions?

- section 3.4: "Conversely, approximately 50% of Smad4KO BRAFV600E/+ non-tumor organoids and almost all tumor organoids survived on tissue culture plastic (Fig 4a)."
According to your figure description it's the other way round - almost all non-tumor organoids and only 50% of tumor organoids survived

- section 3.4/3.5: maybe the reason why you saw only 10 colonies in tumor organoid reexpressing SMAD4 is due to less proliferating cells and cell cycle inhibition in SMAD4 re-expressing cells shown in section 3.5 (figure 6D). 

Thank you very much for that interesting paper. 

Best regards,
